# Transporter-Mediated Cellular Distribution of Tyrosine Kinase Inhibitors as a Potential Resistance Mechanism in Chronic Myeloid Leukemia

**DOI:** 10.3390/pharmaceutics15112535

**Published:** 2023-10-26

**Authors:** Noor E. Verhagen, Jan B. Koenderink, Nicole M. A. Blijlevens, Jeroen J. W. M. Janssen, Frans G. M. Russel

**Affiliations:** 1Division of Pharmacology and Toxicology, Department of Pharmacy, Radboud University Medical Center, 6525 GA Nijmegen, The Netherlands; noor.verhagen@radboudumc.nl (N.E.V.); jan.koenderink@radboudumc.nl (J.B.K.); 2Department of Haematology, Radboud University Medical Center, 6525 GA Nijmegen, The Netherlands; nicole.blijlevens@radboudumc.nl (N.M.A.B.); jeroen.janssen1@radboudumc.nl (J.J.W.M.J.)

**Keywords:** tyrosine kinase inhibitor, drug transporting proteins, ABC transporters, cellular distribution, chronic myeloid leukemia, treatment resistance

## Abstract

Chronic myeloid leukemia (CML) is a hematologic neoplasm characterized by the expression of the BCR::ABL1 oncoprotein, a constitutively active tyrosine kinase, resulting in uncontrolled growth and proliferation of cells in the myeloid lineage. Targeted therapy using tyrosine kinase inhibitors (TKIs) such as imatinib, nilotinib, dasatinib, bosutinib, ponatinib and asciminib has drastically improved the life expectancy of CML patients. However, treatment resistance occurs in 10–20% of CML patients, which is a multifactorial problem that is only partially clarified by the presence of TKI inactivating *BCR::ABL1* mutations. It may also be a consequence of a reduction in cytosolic TKI concentrations in the target cells due to transporter-mediated cellular distribution. This review focuses on drug-transporting proteins in stem cells and progenitor cells involved in the distribution of TKIs approved for the treatment of CML. Special attention will be given to ATP-binding cassette transporters expressed in lysosomes, which may facilitate the extracytosolic sequestration of these compounds.

## 1. Introduction

Chronic myeloid leukemia (CML) is a clonal hematopoietic stem cell (HSC) disorder characterized by a reciprocal translocation between chromosome 9 and chromosome 22. This mutation results in the fusion of the breakpoint cluster region *(BCR)* gene with the Abelson murine leukemia viral oncogene homolog 1 *(ABL1)* [1]. The truncated chromosome 22 containing the *BCR::ABL1* fusion gene is known as the Philadelphia (Ph) chromosome. The Ph chromosome encodes the cytoplasmic localized chimeric BCR::ABL1 oncoprotein, which displays abnormally high and constitutive tyrosine kinase activity [2]. Tyrosine kinases catalyze the transfer of an ATP-derived phosphate group to tyrosine residues on protein substrates. These protein substrates become activated and subsequently trigger various cellular signaling pathways involved in the regulation of cell proliferation and differentiation [3]. The aberrant tyrosine kinase activity of BCR::ABL1 also provides pro-survival signals to malignant cells, which results in increased resistance to apoptosis [4,5]. Ultimately, this causes a massive accumulation of myeloid cells in all stages of maturation in the bone marrow, peripheral blood, and spleen. CML patients suffer from hypermetabolism, fatigue, a loss of appetite, and visual and hearing disturbances. CML has an incidence of 1–2 cases per 100,000 adults and accounts for approximately 15% of newly diagnosed leukemia patients [6]. If inadequately treated or in case of resistance, CML progresses from the chronic phase to a blast phase known as “blast crisis”, which is mostly fatal within 6 months [7]. In the blast phase, CML has all the features of acute myeloid leukemia and results in bone marrow failure.

Imatinib (STI-571, Gleevec) was introduced as a first-line treatment for CML patients in 2001. It is a small molecule that competes for binding to the ATP-binding site of the BCR::ABL1 protein, stabilizing the inactive form of the oncoprotein [8]. Imatinib has proven to be quite effective in comparison to other targeted therapies, with an estimated overall survival rate of 83% at 10 years [9]. Despite the success of imatinib, treatment fails in about half of the patients because of resistance or intolerance, therefore leading to the exploration of other treatment options [10]. Currently, four tyrosine kinase inhibitors (TKIs) are approved for first-line treatment: imatinib, dasatinib, nilotinib, and bosutinib, whereas ponatinib and asciminib are only available in the second or third line of treatment [10,11]. The choice of first-line treatment depends on multiple factors, including individual patient characteristics, preferences, expected drug adherence, lifestyle preferences, comorbidities, distinct drug toxicity profiles, and the experiences of physicians and clinical centers [12].

Resistance to first-line treatment of CML occurs in 10–20% of patients and may be caused by several mechanisms. Pharmacokinetic issues can contribute to lowered cytosolic TKI concentrations, limiting the effectiveness of TKI treatment. Drug–drug interactions that potentially lead to diminished gut absorption, increased metabolism, or drug adherence problems may cause insufficient TKI plasma levels [13]. To achieve satisfactory treatment outcomes, it is critical for TKIs to reach adequate pharmacological concentrations in the cytosol of leukemic hematopoietic stem cells and progenitor cells, where the BCR::ABL1 oncoprotein resides. A key parameter influencing intracellular TKI concentrations is the expression level and activity of drug-transporting proteins [14]. A disturbed balance between transporter-mediated TKI influx and efflux over the cell membrane might explain resistance phenotypes caused by ineffective uptake and/or excessive extrusion, as has been shown with polymorphisms in organic cation transporters [15,16]. Recently, the involvement of transporters in lysosomal accumulation and retention of TKIs has been of increasing interest as an additional potential resistance mechanism [17]. This review addresses the currently available knowledge on transporter-mediated cellular (re)distribution of TKIs in CML stem cells and progenitor cells, with special emphasis on lysosomal transporters.

## 2. CML Stem Cells and Progenitor Cells

Cells from the hematopoietic system are continually regenerated from HSCs residing in the bone marrow. Healthy HSCs are multipotent primitive cells that can develop into all types of blood cells, including myeloid-lineage and lymphoid-lineage cells. They are characterized by their ability to repopulate the bone marrow of irradiated recipients [18,19].

The cells that initiate the ongoing proliferation and expansion of malignant myeloid cells are known as CML leukemic stem cells (LSCs), and they can be found in the CD34+CD38− population [20]. In vitro studies have confirmed that LSCs can remain viable in a quiescent state, even in the presence of growth factors and imatinib [21]. This is in accordance with clinical observations in which CML patients may relapse even after prolonged TKI therapy when BCR::ABL1 has been undetectable for a longer time [22]. Apparently, a population of quiescent LSCs persists in the majority of patients with chronic-phase CML, even in those in deep remission [23,24,25]. This is probably due to the fact that while committed progenitor cells and mature myeloid cells are very sensitive to BCR::ABL1 inhibition by TKIs, this is not the case in quiescent CML stem cells, hereby leaving this population as a disease reservoir [26]. Moreover, it has previously been proposed that TKI therapy itself drives an adaptive ‘quiescence’ response in LSCs by altering their gene expression profile [27]. Single cell analyses have demonstrated an enrichment of LSC populations with quiescent transcriptional signatures during TKI treatment [28,29].

Purifying CML LSCs for in vitro study is not trivial. These cells are located within the CD34+/CD38− cell population but not exclusively. Recently, CD26 (dipeptidyl-peptidase IV) has been identified as a specific CML LSC marker, as it is consistently expressed in chronic-phase CML patients but not in normal HSCs or stem cells of other (myeloid) neoplasms [30].

Remarkably, low levels of *BCR::ABL1* transcripts have been detected in the peripheral blood of otherwise healthy individuals with an increasing frequency upon aging [31,32]. This may imply that BCR::ABL1 is not the only essential factor for the development of CML or suggest that this *BCR::ABL1* expression comes from cells lacking stem cell capacity. Alternatively, it could mean that the transformation of HSCs into LSCs and their ability to function as leukemia-initiating cells is the result of an interplay between intrinsic and extrinsic cellular factors.

## 3. Resistance Mechanisms against TKIs in CML

Resistance to TKI treatment can occur de novo or during treatment and is a complex, multifactorial process that culminates in the selection of a cancer clone able of evading treatment. The development of TKI resistance can be divided into BCR::ABL1-dependent and BCR::ABL1-independent mechanisms [33]. Examples of BCR::ABL1-dependent mechanisms include the overexpression of the BCR::ABL1 oncoprotein and mutations in the TKI-binding domains of the ABL moiety of the BCR::ABL1 fusion protein, which impair the binding of TKIs to their targets [34]. Over 100 *BCR::ABL1* mutations have been detected that may cause acquired TKI resistance, with the most commonly occurring threonine-315 to isoleucine substitution (T315I) remaining difficult to treat [35]. BCR::ABL1-independent resistance is mediated through the activation of alternative survival pathways, such as changes in epigenetics, cellular signaling, the CML microenvironment, and transport protein expression [36,37]. In addition, the main cytochrome P450 (CYP) enzyme, CYP3A4, is involved in the metabolism of almost all TKIs and could also contribute to TKI resistance in leukemia cells [38]. CYP1B1 is expressed in hematopoietic CD34+ stem cells and progenitor cells and is involved in the biotransformation of imatinib [39,40]. The association between TKIs used in the CML treatment and drug metabolizing enzymes is discussed in more detail elsewhere [41].

## 4. Drug Transporter Expression in HSCs and LSCs

The cytosolic TKI concentration and associated BCR::ABL1 inhibition is dependent on the presence of drug-transporting proteins in these cells. In this section, an overview will be given of transporter expression in normal HSCs and CML LSCs.

The organic cation influx transporter OCT1/SLC22A1 has been associated with the uptake of imatinib into leukemic cells [42]. The expression of this transporter is enhanced in the more mature CD34+CD38+ sub-population of hematopoietic cells when compared to their primitive counterparts [43]. This possibly indicates an increase in imatinib influx, and thus, BCR::ABL1 inhibitory potential in this more mature sub-population.

HSCs express several membrane transporters of the ABC superfamily, including P-glycoprotein (P-gp/ABCB1) and breast cancer resistance protein (BCRP/ABCG2) [44,45,46]. Both transporters have previously been shown to play a role in the membrane transport of various endogenous and xenobiotic substrates, including anticancer agents such as TKIs [47,48,49]. P-gp expression has been proven to correlate with an immature immunophenotype in normal hematopoietic cells [45,50]. *ABCB1* mRNA expression in healthy bone marrow cells was higher in immature CD34+CD38− cells compared to more mature CD34+CD38+ cells, an effect that was even more pronounced in CML-derived samples [51,52]. Functional assays, using epifluorescence microscopy in combination with single-cell image analysis, were used to confirm the transporter-mediated efflux of the fluorescent P-gp substrate rhodamine 123 in highly purified viable CD34+CD38− cells [53]. It was also shown that substrate efflux was higher in CD34+CD38− cells compared to more differentiated CD34+CD38+ progenitors [53]. Logically, this would result in more efflux of TKIs that are P-gp substrates in primitive HSCs compared to more mature progenitor cells. In vitro P-gp overexpression was also observed in the CML cell line K562 as an intermediate step during the development of resistance to imatinib [54]. In addition, *ABCG2* mRNA was found at relatively high levels in CD34+CD38− HSCs, but the expression was reduced in more committed CD34+38+ progenitor cells [55]. P-gp protein expression was enhanced in TKI-naïve CML samples compared to healthy bone marrow samples, indicating a greater potential for efflux from leukemic cells before any exposure to TKIs [51]. Less-studied ABC transporters, such as ABCA1, ABCB2, ABCC1, ABCD4, and ABCG1, are also mainly expressed in the immature CD34+CD38− sub-population of HSCs and are down-regulated at the mRNA level in normal human primitive HSCs after differentiation into the CD34+CD38+ sub-population [44,56]. Strikingly, both ABCA1 and ABCG1 are transporters known to be involved in cholesterol efflux. Increased intracellular cholesterol levels promote proliferation and mobilization of HSCs, and thus, these transporters might contribute to their quiescent nature [57]. ABCB2 is known to be involved in the transport of cytosolic proteins into the endoplasmic reticulum [58]. Multidrug resistance protein 1 (MRP1), encoded by the *ABCC1* gene, has been suggested to mediate drug resistance in stem cells. A large number of organic anionic drugs and metabolites, including several anticancer agents, are transported by MRP1 [59,60]. Although in vitro data using the K562 cell line has suggested that MRP1 only plays a minor role in the efflux of imatinib and dasatinib, it would be interesting to investigate whether other TKIs are substrates and whether MRP1 is differentially expressed in chronic phase CML patients [61]. ABCD4 is known to be involved in the transport of vitamin B(12) across liposomal membranes in vitro [62]. *ABCA3* mRNA expression in bone marrow specimens from adult CML patients was significantly upregulated compared to the expression in human bone marrow progenitor cells from healthy volunteers [63]. Expression of *ABCA4*, *ABCA8*, *ABCC9*, and *ABCG4* mRNA was hardly detectable in HSC samples, although it was shown that these transporters are consistently expressed in other types of stem cells, like unrestricted somatic stem cells obtained from cord blood and adult adipose tissue mesenchymal stem cells [64].

Several polymorphic variants of influx and efflux drug transporter genes that result in lower gene expression and altered cellular localization or protein activity have been described, potentially influencing TKI transport and thus the pharmacokinetics of these drugs [65,66,67]. Such variants may influence TKI treatment efficacy but will not be addressed in further detail.

## 5. TKIs and Cellular Transport Mechanisms

Currently, six different TKIs are approved for the treatment of CML (Figure 1). In cases of an insufficient response, the TKI dose can be increased, or a switch to another TKI can be made. Except for asciminib, all TKIs available for the treatment of CML target the ATP-binding pocket of the tyrosine kinase. As this ATP-binding site is highly homologous across the human kinome, these compounds are relatively nonselective. Off-target activities of these compounds, like the inhibition of other kinases besides BCR::ABL1, may contribute to the differences in adverse event profiles observed for different TKIs. For example, it has been shown that the Src family of kinases is not inhibited by imatinib, whereas dasatinib is a potent inhibitor of these kinases [68,69]. Other important kinases like KIT, ARG, and platelet-derived growth factor receptors α and β may also interact with TKIs used in CML treatment and thus cause side effects [70,71]. Asciminib binds to the myristoyl pocket of the ABL1 kinase domain and induces an inactive conformational change in ABL, thereby switching off its enzymatic activity [72]. Conventional TKIs used in the treatment of CML can be divided into type I or type II inhibitors [34]. Type II inhibitors stabilize an inactive enzyme conformation [34]. They exhibit more stringent binding requirements, making them more vulnerable to mutations, but they do have the advantage of increased selectivity [73]. Type I inhibitors recognize the active kinase conformation and compete more directly with ATP for binding [34]. They tend to be more promiscuous but are less prone to the mutational landscape [34]. Imatinib, nilotinib, ponatinib, and asciminib are type II inhibitors [74], dasatinib is a type I inhibitor, and bosutinib exhibits features of both [34].

Interpreting the ATPase assay of ABC transporters is complex because there are transported substrates that inhibit ATPase activity, and substrates that enhance ATPase activity. Furthermore, a TKI may act as a substrate of a particular transporter at low concentrations, while an inhibitory effect can be seen at higher concentrations. Therefore, the compound concentration and binding mode must be considered.

Molecular monitoring of the ratio of *BCR::ABL1* to *ABL1* mRNA transcripts by RT-qPCR is used to assess patient response to TKI therapy and thereby guides clinical patient management [75]. Clinical studies have shown that optimal survival correlates with obtaining a major molecular response, i.e., that the BCR::ABL1 level is ≤0.1%. Patients who attain a stable deep molecular response ≤ 0.01% may attempt to stop TKI therapy. Remarkably, around 40% of patients do not relapse after drug discontinuation, although in many cases, low BCR::ABL1 levels remain detectable [76].

### 5.1. Imatinib

In 2001, imatinib mesylate was the first TKI approved by the FDA and EMA for the treatment of CML. Imatinib is an active site type II inhibitor and is administered in the mesylate form for increased stability and bioavailability [34]. Currently, the recommended therapeutic plasma concentration of imatinib is between 2 and 6 μM, although it has been suggested that this upper limit is too high [77]. Common side effects related to imatinib treatment are nausea, diarrhea, edema, muscle cramps, bone pain, and hematological toxicities [78].

OCT1/SLC22A1 has been reported as the main influx transporter involved in imatinib uptake into CML cells [42]. Higher SLC22A1 mRNA levels have been shown to be predictive of improved patient response to imatinib treatment [79]. On the other hand, patients with low OCT1 activity and trough plasma levels of less than 2.4 μM have inferior outcomes and may benefit from dose intensification [80]. The functional activity of the OCT1 protein in CML patients at diagnosis can be used as a prognostic indicator for both short- and long-term patient response [81]. Although OCT1 is generally accepted as the main influx transporter for imatinib, some studies have reported contradicting results, leaving its role not fully elucidated [82]. For example, OCT1 was shown to be overexpressed in K562 cell lines resistant to imatinib and dasatinib compared to their parental cell line [61]. The authors did not provide an explanation for this rather contradictory result, as decreased expression of influx proteins would be expected in resistant CML cells. Other transporters associated with imatinib influx are OATP1A2, OATP1B3, and OCTN2 [83,84].

The interaction of imatinib with the efflux transporter P-gp has been studied in vitro, and the results indicate that ABCB1/P-gp overexpression is likely associated with the development of imatinib resistance [54,85,86]. Flow cytometry of peripheral blood lymphocytes from 59 CML patients showed a significant increase in P-gp expression in patients who did not respond to imatinib treatment compared to responding patients [87]. In the higher concentration range (>10 μM), imatinib was able to inhibit the ATPase activity of P-gp, indicative of transporter inhibition [88]. The inhibitory potential of imatinib on P-gp is interesting from a mechanistic point of view, but the clinical relevance of this finding is questionable, as these high drug levels may not be achievable in vivo [89].

Multiple studies have reported on the interaction of imatinib with ABCG2/BCRP [46,90,91,92,93]. Some studies have observed imatinib to be a substrate of BCRP [93,94,95], while others suggest that it is an inhibitor [92,96,97]. Recently, cryo-EM has confirmed that imatinib traps the transporter in an inward-facing conformation that ultimately prevents ATP hydrolysis in a similar fashion as the established BCRP inhibitor Ko143 [98]. It has previously been shown that murine Bcrp1 translocation from the membrane to the intracellular compartment is promoted upon treatment with the PI3K inhibitor LY294002 [99]. On the other hand, BCR::ABL1 causes activation of the PI3K/AKT pathway [100]. It has therefore been hypothesized that imatinib inhibition of BCR::ABL1 downregulates PI3K/Akt-signaling, which may, in turn, reduce plasma membrane BCRP activity in CML cells, resulting in less resistance to imatinib [91].

ABCC3/MRP3 mRNA expression was significantly increased in peripheral blood leukocytes of CML patients failing imatinib treatment compared to responders [101]. Additionally, imatinib transport was observed in an MRP3-transfected cell monolayer. This effect could be reversed after addition of the MRP3 inhibitor probenecid [101]. In vitro experiments showed that specific MRP6 inhibitors had no effect on the imatinib IC50, denoting imatinib as an unlikely substrate of this transporter [102].

ABCA3 is known as a lipid transporter involved in the regulation of pulmonary surfactant homeostasis [103]. The protein has been shown to protect leukemic cells from the cytotoxic effects of imatinib. Moreover, transporter expression was enhanced in patient-derived CD34+ cells after exposure to 2 μM imatinib for 48 h in both leukemic and non-transformed hematopoietic stem cells compared to the untreated control, potentially indicating imatinib as an ABCA3 substrate [104].

### 5.2. Nilotinib

In 2007, nilotinib was approved by the FDA and the EMA for CML patients who had developed resistance to or could not tolerate imatinib. Nilotinib, an active site type II inhibitor, is a product of rational drug design based on crystallographic studies on the interaction between imatinib and the ABL kinase domain [105]. The mean trough plasma concentration in CML patients treated with 400 mg of daily nilotinib was 1 μM [106,107]. Nilotinib-mediated antiproliferative activity and inhibition of BCR::ABL1 autophosphorylation in various BCR::ABL1 expressing murine and human cell lines proved to be considerably higher compared to imatinib [108,109]. This finding was clinically confirmed in a phase 3 trial that showed an increased rate of major molecular response at 12 months in newly diagnosed chronic phase Ph+ CML patients treated with nilotinib rather than imatinib, albeit without an improvement in survival rates [110].

In contrast to other TKIs, nilotinib influx is not mediated by OCT1 [42]. In fact, cellular uptake is thought to be predominantly passive, although in vitro experiments using CHO cells transfected with OATP1B1 and OATP1B3 showed increased accumulation of nilotinib, indicating that these proteins may facilitate influx [42,111]. Additionally, nilotinib has been shown to act as an inhibitor of various cationic influx transporters, including OCT1, -2 and -3 [112].

Contradictory results have been reported when it comes to nilotinib as a P-gp/ABCB1 transporter substrate. On the one hand, it was shown that P-gp had only a minor effect on nilotinib cytotoxicity in K562 cells overexpressing the transporter [113]. Also, in MDCKII cells expressing P-gp, no evidence of transport of radiolabeled nilotinib was observed [114]. In this study, nilotinib efflux was not modulated in the presence of the P-gp inhibitors PSC-833, tariquidar, and verapamil [114]. On the other hand, it was shown that nilotinib resistant K562 cells demonstrated up to 4.7-fold greater levels of ABCB1 mRNA compared to control cells and a significant increase in P-gp surface protein expression was observed [54]. At concentrations > 500 nM, nilotinib inhibited the efflux of the prototypic P-gp substrate rhodamine 123 [49,115]. The fact that nilotinib acts as an inhibitor of P-gp in the higher nanomolar range was confirmed by the increased intracellular concentration and sensitivity to chemotherapeutic drugs transported through P-gp when co-incubated with micromolar concentrations of nilotinib [116,117]. This finding could be used as an argument for combinatorial TKI therapy.

Nilotinib is a confirmed BCRP substrate, as in vitro experiments using BCRP-transduced K562 cells demonstrated that overexpression of the transporter has a protective effect against nilotinib-mediated cell death [46,113]. Furthermore, nilotinib competed with the established BCRP substrate [^125^I] IAAP and stimulated the ATPase activity of the transporter [46]. It was also found that nilotinib potentiates the cytotoxicity of widely used BCRP drug substrates such as mitoxantrone and doxorubicin [116].

ABCA3 mRNA induction was observed in CD34+ ABL::BCR1+ leukemic cells after exposure to nilotinib [104]. Although this is not direct proof of nilotinib transport, it is a strong indication that ABCA3 may be involved in the cellular efflux of the drug.

### 5.3. Dasatinib

Dasatinib is an Src inhibitor with ABL inhibitory properties and was thus evaluated for its potential to treat CML [118]. In the Dasision trial, dasatinib showed higher rates of confirmed complete cytogenetic response and major molecular response compared to imatinib treatment, which eventually led to FDA and EMA approval in 2006 for the treatment of CML [119]. After a 5-year follow-up period, the study showed that dasatinib treatment was associated with higher rates of major molecular responses. However, like with nilotinib, no differences were observed in progression-free survival or overall survival between patients treated with imatinib and dasatinib [120]. The peak plasma levels after daily administration of 100 mg dasatinib ranged from 12 to 493 nM [121]. Dasatinib is an active site type I inhibitor and has shown to be highly effective against most of the clinically relevant imatinib-resistant BCR::ABL1 isoforms except for the T315I mutation [122,123].

The intracellular uptake of dasatinib was not significantly reduced when the dual OCT1 and OCT3 inhibitor prazosin was added to mononuclear cells of chronic phase CML patients, indicating that it is not transported by these proteins [124]. Dasatinib uptake into K562 cells was temperature-independent, suggesting a predominantly passive transport mechanism [124]. In K562 and VBL-100 cells overexpressing P-gp, the uptake and retention of dasatinib was significantly lower compared to the respective parental cell lines, indicating a role for this transporter in dasatinib efflux [124]. P-gp overexpression in K562 cells resulted in dasatinib resistance, an effect that could be fully reversed by the addition of the specific P-gp inhibitor PSC833 [113]. Dasatinib inhibited the efflux of the P-gp substrate calcein from these cells at concentrations >10 μM [113]. As therapeutic plasma concentrations are much lower, the clinical relevance of dasatinib-mediated P-gp inhibition seems limited.

Dasatinib is transported by BCRP, as shown by decreased intracellular concentrations in BCRP-overexpressing cell lines K562-ABCG2 and Mef-BCRP1 compared to their parental cell lines [113]. In the K562-ABCG2 cell line, the uptake and retention were significantly increased with the addition of the BCRP inhibitor Ko143, an effect that was not observed in the parental K562 cell line [124].

ABCA3 expression was induced in CD34+ ABL::BCR1+ leukemic cells upon exposure to dasatinib [104]. Taken together with the observation that ABCA3 overexpression induces a significant reduction in susceptibility to cytotoxic agents like daunorubicin, mitoxantrone, and vincristine, it is likely that dasatinib is a substrate of the transporter, although direct evidence is lacking [125].

### 5.4. Ponatinib

Ponatinib was approved by the FDA in 2012 but was shortly thereafter withdrawn due to serious cardiovascular safety concerns. It returned to the market in 2013 with an indication limited to patients with a T315I mutation or for whom no other TKI therapy was indicated, with revised warnings and precautions [126]. Ponatinib is a third-generation type II inhibitor designed to overcome the gatekeeper T315I mutation, which confers resistance to imatinib, dasatinib, nilotinib, and bosutinib [127]. The mean steady-state peak plasma concentration reached after a once-daily oral dose of 45 mg in patients with advanced hematologic malignancies was 137 nM [126]. A phase 2 trial investigated the effect of ponatinib on heavily pretreated CML patients who had either unacceptable side effects from dasatinib or nilotinib or who had the BCR::ABL1 T315I mutation. A major cytogenetic response, i.e., no more than 35% of the cells in the bone marrow contain the Ph chromosome, was seen in 51% of the patients with unacceptable side effects from dasatinib or nilotinib and in 70% of the patients with a T315I mutation [128].

To establish whether ponatinib is taken up by OCT1-3, in vitro experiments with the inhibitors prazosin, procainamide, and ibuprofen were performed in K562 cells. Inhibition of the transporters had no effect on ponatinib-mediated cytotoxicity, indicating that it is not a substrate [129].

Unlike most other TKIs, ponatinib is not transported by P-gp and BCRP [129]. Nevertheless, the interaction of ponatinib with these proteins turned out to induce a significant concentration-dependent accumulation of P-gp and BCRP substrates in K562 cells overexpressing these efflux transporters [130]. Moreover, ponatinib acted synergistically with chemotherapeutic drugs that are BCRP substrates by inducing cytotoxicity and apoptosis in 8226/MR20 cells overexpressing BCRP [130]. It can thus be concluded that ponatinib is an inhibitor of BCRP and P-gp.

Ponatinib was also shown to interact with MRP7, a protein encoded by the ABCC10 gene [131]. Intracellular accumulation of the MRP7 substrate paclitaxel was significantly enhanced in MRP7-expressing human embryonic kidney (HEK) 293 cells upon ponatinib treatment [131]. This effect was facilitated by the inhibition of the MRP7 transporter function and downregulation of the protein in a time- and concentration-dependent manner [131]. Whether ponatinib is solely an inhibitor, or whether it is also transported by MRP7, remains to be elucidated.

### 5.5. Bosutinib

Bosutinib is a second-generation active site TKI approved by the FDA in 2012 and by the EMA in 2013 for treatment of CML patients. In vitro experiments have shown that bosutinib has antiproliferative activity against multiple BCR::ABL1-positive leukemia cell lines as well as the ability to block BCR::ABL1 phosphorylation at sub-micromolar concentrations [132]. The mean trough and peak levels in newly diagnosed chronic phase CML patients within 3 months after the administration of 500 mg/day of bosutinib were 294 nM and 456 nM, respectively [133]. Bosutinib appeared to be effective in both imatinib-resistant and -intolerant patients across all BCR::ABL1 mutations except for the T315I mutation [134]. The drug has the important advantage that it is not efficiently extruded by multidrug resistance transporters [135]. In the BFORE trial, the efficacy and safety of bosutinib versus imatinib in the first-line treatment of chronic phase CML was assessed [136]. The total major molecular response rate at 5 years was significantly higher with bosutinib than with imatinib (74% vs. 65%, respectively) [136]. It was concluded that bosutinib is effective and has an acceptable safety profile that is different from that of other TKIs, which might offer a new treatment option for patients with chronic phase CML who are resistant or intolerant to imatinib [136].

Passive uptake is thought to play an important role in the cellular distribution of bosutinib as it is a relatively hydrophobic compound. Bosutinib has been proposed as an OCT2 inhibitor because serum levels of creatinine, a known OCT2 substrate, were elevated in CML patients treated with the drug [137].

Conflicting results have been reported when it comes to the interaction between bosutinib and P-gp. The cellular concentration of bosutinib was significantly lower in P-gp-overexpressing K562 cells compared to the parental cell line [138]. Moreover, BCR::ABL1 phosphorylation was reduced by dual treatment with the P-gp inhibitor verapamil and bosutinib [138]. However, another study showed that the P-gp overexpression in K562 cells had no significant effect on its cellular toxicity [113]. The efflux of the fluorescent P-gp substrate Hoechst 33343 was inhibited in a concentration-dependent manner when high concentrations of bosutinib were added [113]. Whether bosutinib is transported by P-gp therefore remains an open question.

BCRP overexpression in K562 cells showed a minor protective effect against bosutinib treatment, although this did not result in altered intracellular bosutinib levels [113]. High micromolar concentrations of bosutinib were shown to inhibit the BCRP ATPase activity [113].

### 5.6. Asciminib

In October 2021, asciminib was approved by the FDA for the treatment of chronic phase CML patients with resistance or intolerance to two prior lines of TKI therapy and for patients with a T315I mutation [11]. A daily dose of 40 mg asciminib resulted in a maximum plasma concentration of 765 nM in patients [139]. Asciminib differs from previously approved ABL1 kinase inhibitors in that it does not bind to the ATP-binding pocket of the kinase. Instead, the compound acts as an allosteric inhibitor and engages a vacant pocket at a site of the kinase domain normally occupied by the myristoylated N-terminal of ABL1. This motif serves as an allosteric negative regulatory element but is lost on fusion of ABL1 to BCR [140]. The binding of asciminib in this pocket mimics the binding of myristate and thus stabilizes the assembled inactive state of the ABL kinase [140]. Because of the distinct conformation of the myristoyl pocket, the selectivity of asciminib for ABL1 is high [141]. Additionally, it results in asciminib activity against ATP-binding site mutated forms of the BCR::ABL1 kinase [141]. In vitro experiments with BCR::ABL1-transfected cells showed potent asciminib-mediated inhibition of proliferation [142]. In addition, synergistic apoptosis-inducing effects of asciminib and ponatinib were observed in CD34+/CD38− CML stem cells obtained from chronic phase and blast phase patients with the T315I gatekeeper mutation [143]. The combination of asciminib with nilotinib and asciminib with imatinib resulted in increased cytotoxicity in both parental and asciminib-resistant K562 cells [144]. This could be due to competitive inhibition of BCRP as in vitro studies have suggested that asciminib is a BCRP substrate [145,146]. An alternative explanation may be that the simultaneous targeting of the myristate binding pocket as well as the ATP-binding pocket is more effective than targeting either site alone [144].

Asciminib-mediated cell death was significantly decreased in P-gp-overexpressing K562 cells compared to their parental cells, an effect that was nullified by the addition of the P-gp inhibitor cyclosporine [144,145]. Upon exposure to asciminib the P-gp overexpressing cell line K562-Dox showed a 2.1-fold further increase in P-gp protein expression as compared to control [144]. These findings are indicative of P-gp -mediated asciminib transport.

Research regarding the influence of drug transporting proteins involved in asciminib distribution is still relatively scarce, due to its recent introduction. Further studies are required to elucidate whether other efflux proteins are also involved.

## 6. Lysosomal TKI Sequestration

Until now, most research on the relationship between drug-transporting proteins and TKIs has focused on transporters located in the plasma membrane. However, lipophilic cationic drugs like TKIs tend to accumulate in lysosomes, a process known as lysosomal trapping [147]. Drug transporters expressed in lysosomes might also contribute to the reduction in TKI effectiveness by limiting their access to the target site in the cytosolic compartment. Lysosomes are acidic organelles that have been suggested to play a role in the accumulation of hydrophobic weak-base anticancer drugs such as TKIs [17,148]. Once these drugs encounter the acidic pH of the lysosomal lumen, they become protonated and are no longer able to diffuse across the lysosomal membrane [149].

Various ABC transporters, including ABCA2, ABCA3, ABCA5, ABCB6, ABCB9, and ABCD4, are present in the lysosomal membrane (Figure 2) [125,150]. Given their membrane orientation in which the ATP-binding domain faces the cytoplasm, an influx function is predicted, implying the uptake of substrates into the lysosomes [150]. This process could further contribute to the lowering of cytosolic TKI concentrations and thus the ineffectiveness of the TKIs. The contribution of ABCA3 to lysosomal imatinib sequestration has been demonstrated in CML cell lines [151]. Imatinib storage capacity was 2.9 times higher in isolated ABCA3-expressing lysosomes compared to ABCA3-negative lysosomes [151]. In addition, increased expression of ABCA3 has been associated with an unfavorable treatment outcome in acute myeloid leukemia [63]. Whether ABCA3 expression is also associated with poor efficacy of TKIs in CML remains to be investigated. P-gp/ABCB1 has been localized to various intracellular compartments, including lysosomes [150]. Whether the transporter is functional or merely present as digested fragments in the lysosomes is not yet clear [152].

## 7. Discussion

The development of TKIs has revolutionized CML therapy in a way that the life expectancy of patients is approaching that of the general population [153]. This has resulted in a rising prevalence of CML and thus increasing numbers of patients who are dependent on TKI treatment. The effectiveness of the six orally administered TKIs in CML is largely dependent on their cytosolic concentration in CML cells. Drug-transporting proteins expressed at both the plasma membrane and lysosomes influence the cytosolic TKI concentration and may thus play a role in resistance to TKI therapy. This review article gives an overview of current knowledge on the involvement of drug-transporting proteins in the distribution of TKIs used for CML treatment.

In this review, different types of studies have been included, in which a distinction can be made between studies that directly prove the interplay between drug-transporting proteins and TKIs and studies that provide more indirect types of evidence, such as an improved patient response with higher transporter expression (Table 1). The latter type does give a strong indication of the involvement of the drug-transporting protein in question but does not provide a definitive answer to the question of whether the TKI is a substrate of the transporter. A wide variety of influx and efflux transporters are involved in TKI distribution, and some TKIs can also act as inhibitors of these transporters. The question remains whether the influence of transporter-mediated cellular distribution of TKIs can significantly impact patient treatment. A previous study using a CML mouse model has shown that the loss of Abcb1a/b expression in HSCs does not improve the imatinib response [154]. Nevertheless, certain single nucleotide polymorphisms in drug-transporting proteins have been associated with altered response rates to TKI therapy [155,156]. Further research is thus warranted to elucidate the magnitude of the contribution of drug transporters to TKI resistance in CML patients. This is especially true for the more recently approved TKIs, as their interaction with drug-transporting proteins has not been studied as extensively.

In vitro studies have previously demonstrated the effectiveness of P-gp inhibitors in reversing imatinib resistance, suggesting that these compounds could function as TKI-sensitizers [157,158]. However, the systemic expression of ABC transporters makes the translation of in vitro results challenging in the clinical setting. The use of these inhibitors could lead to unexpected organ-specific toxicity or affect the pharmacokinetics of co-administered drugs [15]. Up until now, chemo-sensitizers designed to inhibit drug transporters in vivo have been unsuccessful [159]. Whether inhibitors of drug-transporting proteins could be a feasible treatment option in CML patients should be thoroughly evaluated in clinical trials.

Some TKIs are substrates for certain drug-transporting proteins, whereas other TKIs have an inhibitory effect on the same proteins. This implies that if co-administered, these TKIs could have an additive positive effect in CML treatment. Currently, multiple clinical trials are evaluating the feasibility of combining asciminib with other TKIs used in CML treatment, as asciminib targets a different region on the BCR::ABL1 oncoprotein [160].

Even with the current success rate of TKIs, there is still a considerable number of patients who do not attain optimal responses and are thereby at risk of developing a blast crisis. Allogeneic stem cell transplantation is a last resort for multi-TKI-resistance in CML, but this is often not feasible due to unfitness of the patient because of age or comorbidities. Optimizing treatment outcomes for CML patients, potentially involving the modulation of transport proteins, therefore remains an unmet medical need. A better understanding of the role of drug transporters in TKI distribution can aid in developing more effective and personalized treatment options. Since multiple first-line options are currently available in the treatment of CML, it can be envisioned that the patient-specific transporter expression profile could be taken into account in the choice of the TKI [161]. Insight into the transport mechanisms of TKIs in leukemic hematopoietic stem cells and progenitor cells may also contribute to the drug discovery process by designing molecules with less affinity for efflux transporters and a lower risk of developing resistance.

## Figures and Tables

**Figure 1 pharmaceutics-15-02535-f001:**
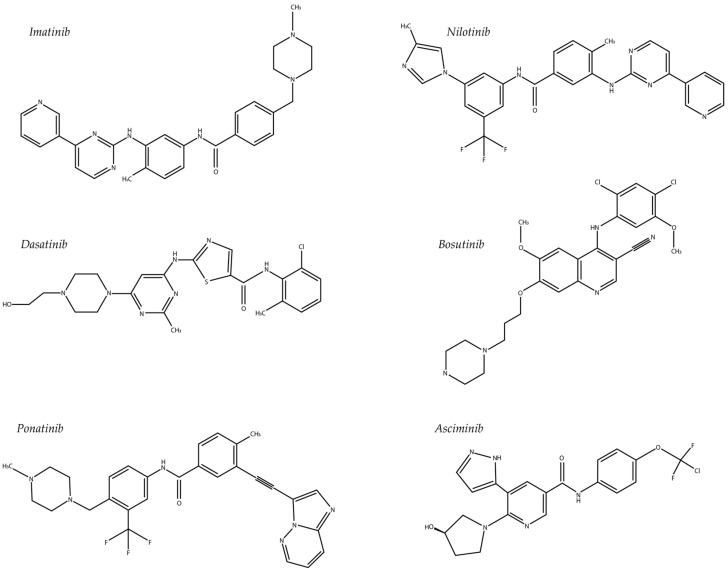
TKIs currently approved for CML treatment.

**Figure 2 pharmaceutics-15-02535-f002:**
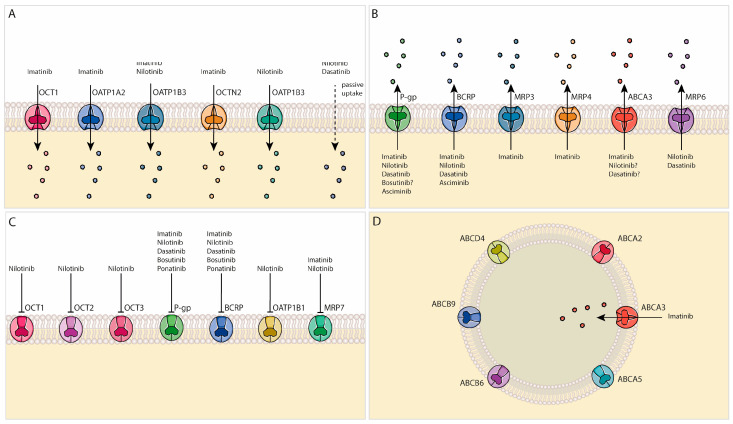
Overview of the currently available data on transporter-mediated distribution of TKIs in leukemic stem cells discussed in this review; (**A**) cellular TKI uptake; (**B**) transporter-mediated TKI efflux; (**C**) inhibition of transporter activity by TKIs; (**D**) lysosomal expression of drug transporters and their impact on TKI transport.

**Table 1 pharmaceutics-15-02535-t001:** Overview of the current knowledge regarding drug-transporter-mediated TKI efflux. Direct evidence is defined as assays where overexpression of a drug transporting protein resulted in reduced intracellular TKI concentrations; all other assay types are listed as indirect evidence.

Compound	Drug Transporting Protein	Direct/Indirect Evidence	Readout	Source
Imatinib	P-gp	Indirect	Increased intracellular uptake and retention in P-gp overexpressing LLC-PK1 cells upon addition of the specific P-gp inhibitor cyclosporin A.	[85]
	BCRP	Direct	Decreased imatinib uptake in BCRP overexpressing cell lines and increased intracellular uptake and retention upon addition of the specific BCRP inhibitor Ko-143.	[93]
	ABCC3	Direct	Increased efflux in ABCC3 overexpressing MDCKII cell monolayers that could be nullified by the addition of the ABCC3 inhibitor probenecid.	[101]
	ABCA3	Indirect	Enhanced transporter expression in CD34+ BCR::ABL1+ leukemic cells after imatinib exposure.	[104]
Nilotinib	P-gp	Indirect	No evidence of radiolabeled nilotinib efflux in P-gp overexpressing MDCKII cells, but significant upregulation of P-gp expression in nilotinib resistant K562 cells.	[54,114]
	BCRP	Indirect	BCRP overexpression in K562 cells protects against nilotinib-mediated cell death.	[46,113]
	ABCA3	Indirect	Enhanced transporter expression in CD34+ BCR::ABL1+ leukemic cells after nilotinib exposure.	[104]
Dasatinib	P-gp	Direct	Reduced intracellular uptake and retention of dasatinib in P-gp overexpressing K562 cells, which could be reversed by a specific P-gp inhibitor.	[113,124]
	BCRP	Direct	Reduced intracellular uptake and retention of dasatinib in BCRP overexpressing K562 cells, which could be reversed by a specific BCRP inhibitor.	[113,124]
	ABCA3	Indirect	Enhanced transporter expression in CD34+ BCR::ABL1 leukemic cells after dasatinib exposure.	[104]
Bosutinib	P-gp	Indirect	Lower intracellular uptake and retention of bosutinib in P-gp overexpressing K562 cells and reduced BCR::ABL1 phosphorylation upon co-treatment of bosutinib and a specific P-gp inhibitor. However, P-gp overexpression had no effect on bosutinib-mediated cellular toxicity.	[113,138]
	BCRP	Indirect	Minor protective effect against bosutinib treatment in BCRP overexpressing K562 cells.	[113]
Asciminib	P-gp	Indirect	Decreased asciminib-mediated cell death in P-gp overexpressing K562 cells, which was nullified upon inhibition of P-gp.	[144,145]

## Data Availability

Data sharing not applicable.

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
