# Peer review of "Transporter-Mediated Cellular Distribution of Tyrosine Kinase Inhibitors as a Potential Resistance Mechanism in Chronic Myeloid Leukemia"

_pharmaceutics, 2023, doi:10.3390/pharmaceutics15112535_

Round 1

Reviewer 1 Report

Comments and Suggestions for Authors

In this review article, the authors summarize the knowledge on TKI agents in CML disease with respect to cell membrane transporters and lysosomal transporters. The article is a thorough and comprehensive summary of the topic, citing both relevant and recent studies. I recommend its publication. I have only a few minor comments below. I would suggest some additions and minor corrections to the final version.

Reviewer 2 Report

Comments and Suggestions for Authors

The authors present a nice summary about the role of drug transporters for TKI resistance. Much work has been done to unravel the complex interplay of membrane-located transporters, drug uptake and functional effects of TKI. The authors now stress another important aspect: expression and activity of drug transporters at the lysosome, trapping TKI drugs into this organelle, and thus mediating TKI resistance.

While this topic is interesting and potentially meaningful, the authors should revise the manuscript by considering the following points:

1. In the Introduction, the authors say PK issues can lead to TKI resistance. This is misleadingly phrased. Rather, PK issues can lead to TKI ineffectiveness given the lowered plasma levels. But the target cells are still sensitive at this moment (not TKI resistant).

2. Although this review deals with drug transporters, the authors should briefly ellaborate on drug metabolism-mediated TKI resistance in leukemia cells. Please present the current evidence of such a cellular resistance mechanims. Do leukemia cells express high CYP3A activity and thus metabolitze TKI extensively?

3. The authors present many examples of the association (and correlation) of drug transporter expression and TKI resistance (in vitro or in vivo). However, it remains open whether this is a coincidence by chance or whether the overexpression really is the cause of TKI resistance. Accordingl,y the authors should present definite evidence that drug transporters lower the intracellular concentration and that this leads to TKI resistance. To do this transparently and comprehensively, the authors should create a table that lists this evidence. To save space, figure 1 can be ommitted instead because it presents text book knowledge only.

Comments on the Quality of English Language

Please work on grammar, punctuation (false comma), and wording. In English, we do not combine words such as "stem and progenitor cells". This likely is Dutrch or German. It is stem cells and progenitor cells.
